# Using the Immunophenotype to Predict Response to Biologic Drugs in Rheumatoid Arthritis

**DOI:** 10.3390/jpm9040046

**Published:** 2019-10-02

**Authors:** Ben Mulhearn, Anne Barton, Sebastien Viatte

**Affiliations:** 1Manchester Collaborative Centre for Inflammation Research (MCCIR), Division of Infection, Immunity and Respiratory Medicine, School of Biological Sciences, Faculty of Biology, Medicine and Health, Manchester Academic Health Science Centre, University of Manchester, Manchester M13 9NT, UK; sebastien.viatte@manchester.ac.uk; 2Centre for Genetics and Genomics Versus Arthritis, Division of Musculoskeletal and Dermatological Sciences, School of Biological Sciences, Faculty of Biology, Medicine and Health, Manchester Academic Health Science Centre, University of Manchester, Manchester M13 9PT, UK; anne.barton@manchester.ac.uk; 3Kellgren Centre for Rheumatology, NIHR Manchester BRC, Manchester University NHS Foundation Trust, Manchester Academic Health Science Centre, Oxford Road, Manchester M13 9WL, UK; 4Lydia Becker Institute of Immunology and Inflammation, Division of Musculoskeletal and Dermatological Sciences, School of Biological Sciences, Faculty of Biology, Medicine and Health, The University of Manchester, Manchester M13 9NT, UK

**Keywords:** immunology, rheumatoid arthritis, T cells, cytokines, biologic drugs, precision medicine, biomarkers

## Abstract

Tumour necrosis factor (TNF)-α is a key mediator of inflammation in rheumatoid arthritis, and its discovery led to the development of highly successful anti-TNF therapy. Subsequently, other biologic drugs targeting immune pathways, namely interleukin-6 blockade, B cell depletion, and T cell co-stimulation blockade, have been developed. Not all patients respond to a biologic drug, leading to a knowledge gap between biologic therapies available and the confident prediction of response. So far, genetic studies have failed to uncover clinically informative biomarkers to predict response. Given that the targets of biologics are immune pathways, immunological study has become all the more pertinent. Furthermore, advances in single-cell technology have enabled the characterization of many leucocyte subsets. Studying the blood immunophenotype may therefore, define biomarker profiles relevant to each individual patient’s disease and treatment outcome. This review summarises our current understanding of how immune biomarkers might be able to predict treatment response to biologic drugs.

## 1. Introduction

Rheumatoid arthritis (RA) is a highly heterogeneous autoimmune disease characterised by inflammation of synovial joints with a prevalence of 0.5–1% and an incidence of 20–50 per 100,000 annually [1]. Inflammation in the joints causes stiffness, pain, and swelling and may eventually lead to joint destruction due to cartilage and bone destruction. The heterogeneity of RA is becoming clearer as more evidence supports that the diagnosis encompasses a number of genetically related diseases that share joint inflammation as the presenting feature [2].

As a complex disease, subtle differences will also exist in the underlying immunological aetiopathologies. Current treatment strategies are designed to standardise treatment across patient groups which superficially appear similar but might have very different disease processes. Control of inflammation is critically important in the treatment of RA because there is a window of opportunity between the onset of inflammation and the onset of structural joint damage [3]. Structural joint damage leads to permanent disability which is detrimental to both the individual and societally [4]. It may even be possible to shut down the disease process entirely in very early rheumatoid arthritis (VERA), which is defined as less than three months of disease activity [5]. Personalised medicine aims to break the ‘one size fits all’ approach by tailoring therapy to subgroups of individuals with the aim of achieving a higher degree of treatment success from the outset.

The personalised approach will likely involve clinical, genetic, immunological, and hitherto unknown factors to inform treatment choice. Given that all treatments for RA without exception, target the immune system, whether that be by modulating serum cytokine signaling or by altering the cellular landscape, logic dictates that an understanding of the immune landscape before treatment begins should inform the prediction of treatment success or failure. This review aims to summarise our current understanding of how immunology might inform treatment choice in RA.

## 2. Current Biologic Therapies Available

There are four main classes of biologic drugs available in the United Kingdom, namely TNF inhibition, IL-6 inhibition, co-stimulation blockade, and B cell depletion. Newer targeted therapies such as JAK/STAT pathway inhibitors (e.g., tofacitinib and baricitinib) are not biologic drugs but act through tyrosine kinase inhibition and are known as ‘small molecule inhibitors’ (SMIs). In practice, as long as there are no contra-indications, the first-choice biologic/targeted therapy is usually the cheapest agent negotiated by the local Clinical Commissioning Group. In the UK, that is still currently anti-TNF biologics, but the situation may change according to cost, and biosimilars which are the biologic equivalent of generic medicines, are continually driving down the cost of biologic drugs. Rituximab as a first-line agent is reserved for those who have lupus cross-over syndrome, Felty’s syndrome, or where anti-TNF is contra-indicated, but it currently remains a second-line agent for use after anti-TNF failure. Although it is recommended that all biologics are prescribed alongside methotrexate or another conventional disease-modifying anti-rheumatic drug (DMARD), adalimumab, etanercept, certolizumab pegol, and tocilizumab are licensed for use as monotherapy [6]. Figure 1 shows the sites of action of the main biologics used in RA.

### 2.1. Anti-TNFs

At present, available anti-TNF therapies include etanercept, adalimumab, infliximab, golimumab, and certolizumab pegol. Broadly, all are equally efficacious as each other, although for reasons not yet fully understood, some work better than others on an individual basis. All, except etanercept, are monoclonal antibodies, whilst etanercept consists of the TNF receptor fused to an Fc domain of IgG1 which binds free TNF [7]. The formation of anti-drug antibodies (ADAs) is one of the reasons for the secondary failure of anti-TNF monoclonal antibody drugs [8,9].

### 2.2. Tocilizumab

IL-6 was first found to be an important cytokine in RA as it drives the acute phase response causing systemic inflammation, including fever, high C-reactive peptide (CRP), and fatigue. Tocilizumab, a monoclonal antibody against the IL-6 receptor, blocks the actions of IL-6 and is effective in RA [10]. Tocilizumab monotherapy also reduces the radiographic progression of erosions in those with highly active disease [11]. It also improves the anaemia of chronic disease found in active RA compared to other biologics, DMARDs, and tofacitinib [12].

### 2.3. Abatacept

Cytotoxic T-lymphocyte-associated protein 4 (CTLA4) is a potent molecule found on activated T cells and Tregs, and binds the co-stimulatory molecules CD80 and CD86 found on antigen-presenting cells (APCs) with greater affinity than their ligand CD28, which is required for T cell receptor (TCR) activation on T cells (reviewed in [13]). The net effect is that naïve T cell activation is abrogated and CD80/CD86 expression on APCs is downregulated. Abatacept, a CTLA4-Fc fusion protein, was found to be efficacious for RA after inadequate response to an anti-TNF in the ATTAIN trial [14]. It is more expensive, although can be used first-line where anti-TNF agents might be contra-indicated. It appears to cause fewer serious adverse effects including infections than the other biologics as found in a Cochrane review [15].

### 2.4. Rituximab

Rituximab is an anti-CD20 antibody that depletes B cells and is effective in RA, further supporting a key role for B cells in RA pathogenesis [16]. The REFLEX trial found long-lasting response after one course of rituximab in those with inadequate response to one or more anti-TNFs [17]. Rituximab is effective in 70% of seropositive and 48% of seronegative patients [18]. The main safety considerations are the development of hypogammaglobulinaemia and neutropaenia which predispose to serious infections, and the reactivation of a number of fatal brain infections such as JC virus [19].

## 3. Predicting Response to Biologic Drugs

Predicting response to therapy is not currently possible using any meaningful demographic or biological parameters available in the clinic. It is however, an active area of research and highlighted below are some of the methods research groups have tried to address this problem. There are some promising results, although to date none of these have been translated into clinical practice. This is likely due to a lack of validation across different RA cohorts, low positive predictive values, and the unavailability in clinical practice of some of the biomarkers used, such as gene expression profiling or immunological assays.

### 3.1. Clinical and Demographic Predictors of Response

Baseline characteristics can predict better or poorer treatment outcomes. Response to biologics is associated with male gender and concomitant methotrexate, whereas non-response is associated with smoking and having a high baseline health assessment questionnaire (HAQ) score [20]. A regression model using clinical predictors of golimumab found that male gender, younger age, lower health assessment questionnaire (HAQ), erythrocyte sedimentation rate (ESR)/C-reactive peptide (CRP), tender joint count (TJC)/swollen joint count (SJC), and the absence of co-morbidities, could estimate remission rate using 3000 real-world patients [21]. This approach found that those with no comorbidities and the lowest inflammation were more likely to respond, therefore not being particularly informative in terms of choosing a particular biologic drug. Machine learning approaches have also been employed to generate treatment response algorithms. Miyoshi et al. (2016) reported that infliximab response could be predicted by using the nine variables of ESR, TJC, albumin, monocyte count, red blood cell number, prednisolone dose, methotrexate dose, HbA1c, and previous biologic exposure with 92% accuracy [22]. However, these results were not confirmed in other anti-TNF cohorts, and infliximab is rarely used in the UK because it has to be given intravenously compared to the subcutaneous route of other anti-TNFs.

### 3.2. Immunological Predictors of Response

Interest has grown in identifying immunological biomarkers of treatment response. Methods include proteomic analysis of serum and flow cytometry to analyse panels of cell surface and intracellular markers of peripheral blood mononuclear cells (PBMCs). Synovial tissue and fluid are other sources of identifying biomarkers that may not be found in peripheral blood in the same quantities, given that the joint is the main site of pathology in RA. However, peripheral blood from patients is readily accessible in the clinic and PBMCs can be used in immunological studies and stimulation assays, whereas serum can be used for proteomics. The following section reviews immune biomarkers at baseline which have been reported to predict the response to biologics. Table 1 shows a summary of the main studies investigating immune biomarkers as predictors of treatment response.

#### 3.2.1. Anti-Citrullinated Peptide Antibodies and Rheumatoid Factor

Many groups have investigated rheumatoid factor (RF) and anti-citrullinated peptide antibody (ACPA) positivity to predict response to therapy of all biologics. The reports for anti-TNF therapy are either conflicting [23,24,25,29,30] or did not find a correlation [26,27]. A meta-analysis of 5561 patients also did not find an association between ACPA/RF status and anti-TNF response [28].

Abatacept response appears to correlate with ACPA positivity with an odds ratio (OR) estimated to be between 1.4 and 1.9 in multiple studies [31,32,33]. The greatest response rates were seen in those with the highest titers of ACPA [29]. One real-world study found double ACPA/RF positivity resulted in higher abatacept retention rates, suggesting the efficacy of the drug in this multicenter cohort [34].

High titers of ACPA also predicted the best response to rituximab with an OR of 5.1 for good European League Against Rheumatism (EULAR) response [37]. A meta-analysis of 2177 patients confirmed the association between seropositivity and rituximab response, however, the effect was modest [36]. Finally, in another meta-analysis, RF status was associated with rituximab and tocilizumab response but not with abatacept [35].

#### 3.2.2. Serum Biomarkers

Serum biomarkers are easy to measure and have been extensively researched because biologic drugs have cytokines as their target. Some promising reports are outlined below, although none have been validated in independent cohorts to date.

IL-6 at baseline is higher in responders to both etanercept and tocilizumab [38,39]. Furthermore, Shi et al. (2017) noted that high serum IL-6 and low serum survivin at baseline was associated with etanercept response with an OR of nearly 20 [38], and Diaz-Torne et al. (2017) found the patients showing the greatest response to tocilizumab had a combination of high serum IL-6 and low serum soluble IL-6 receptor [39].

IL-33 is a Th2 polarising cytokine, and when detected and added to ACPA status and serum IgG level, predicted 100% of rituximab responders with an OR of almost 30 in one study [40], although this work is yet to be validated in an independent cohort.

Chemokines direct the migration of leucocytes along a concentration gradient and are important to home cells to the sites of inflammation. Receiver operating characteristic (ROC) curve analysis found that high pre-treatment titers of the serum chemokines CXCL10 and CXCL13 predicted response to adalimumab and etanercept with an area under the curve (AUC) of 0.83 [41]. CCL19 was modestly associated with response to Rituximab with an OR of 1.48 [43]. In an innovative study integrating histopathology of synovial tissue with transcription profiling and serology, Dennis et al. (2014) found that elevated soluble ICAM1 (sICAM1) in the serum was associated with a synovial myeloid cell pathotype and good response to adalimumab, whereas CXCL13 predicted a synovial lymphoid pathotype and correlated with good tocilizumab response [42].

Proteomic approaches have also been tested for all the biologics, and although serum changes are usually seen in responders after treatment has begun, no baseline signature has yet been detected by most [56,57,58]. Obry et al. (2015) used quantitative mass spectrometry and identified 12 biomarkers which may have the capacity to predict response to etanercept/methotrexate [59]. This study found that S100A9 protein had the largest influence, although these findings might not be specific to anti-TNF treatment.

#### 3.2.3. Adaptive Immune Cells

Given that the adaptive immune system is thought to be pivotal in the development and perpetuation of autoimmune diseases, investigating these populations in blood and at sites of inflammation could reveal important treatment response biomarkers. There is, however, a lack of consistency in studies on T and B cells in pre-treatment samples for predicting subsequent response.

Baseline high levels of CD27^+^ memory B cells were associated with treatment response to anti-TNFs in one study with a relative risk (RR) of 4.9 [44]. High baseline memory B cells may also be associated with a good response to abatacept [45]. For rituximab, increased pre-treatment CD27^−^ B cells (alongside RF positivity and a normal B cell count) were associated with response in one study [46].

Regarding T cells, CD8^+^ T cells specific for apoptosis-related antigens seen in chronic inflammation were significantly elevated in responders to anti-TNF therapy [47]. Baseline CD8^+^CD28^−^ T cells were associated with a four-fold probability of response to abatacept at six months [48].

#### 3.2.4. Innate Immune Cells

Monocytes can be subdivided into three groups according to their expression of CD14 and CD16 [60]. In RA, there is an increase in the number of CD16^+^ monocytes [61]. Chara et al., (2015) found that high pre-treatment levels of CD14^+^CD16^−^ and CD14^+^CD16^+^ monocytes correlated with reduced response to methotrexate [62]. The same group previously showed that these monocyte subsets remain high in non-responders to adalimumab/methotrexate after three months of therapy with 86% positive predictive value (PPV) [63]. Monocyte subsets might therefore, prove to be an attractive pre-treatment biomarker, and further research should aim to uncover if these cell numbers change in response to other treatment regimens.

Another group studied NK cells and found that low CD56^bri^CD16^-^ NK cells at baseline were associated with response to tocilizumab, but not to anti-TNF therapy [49]. CD56^dim^CD16^+^ NK cells showed no association, thereby leading the authors to conclude that there may be a pathogenetic link between patients who respond to anti-IL-6 therapy and their CD56^bri^CD16^−^ NK cells. Interestingly, such NK cells are capable of secreting many cytokines including GM-CSF, IFNγ and TNF [64].

#### 3.2.5. Interferon Gene Signatures

The interferon gene signature refers to type I interferons and includes genes up and down-regulated by interferons-α and -β. It is often measured indirectly by interferon-response gene quantification where a score is then assigned. In RA patients, 50% with established disease have a high type I interferon signature, and some hypothesise that these patients may have different underlying mechanisms of disease [65]. A high baseline interferon signature was significantly associated with anti-TNF response [50]. In contrast, detecting a high type I interferon signature may predict reduced response to rituximab [51]. This was supported by a prospective analysis of 14 RA patients commenced on rituximab, with further validation in 26 RA patients with an area under the curve (AUC) of 0.87 [52]. Furthermore, prednisolone treatment drives down type I interferon signatures and prediction of rituximab response was therefore, highest in the prednisolone-negative group with an AUC of 0.97 [66]. These data suggest that those patients with a high baseline type I interferon signature respond well to anti-TNF and less well to rituximab, which may indicate differences in pathogenesis in the two groups. However, there is significant methodological variability over different studies in detecting the type I interferon signature which means its use as a clinical biomarker is limited until a standardised approach is developed [67].

### 3.3. Multiplexed Prediction Models

It is likely that a combination of demographic, clinical, laboratory markers, genetics, epigenetics, as well as proteomics, functional immunology, and gene expression profiling will be needed to build up a personalised profile to predict treatment response for each individual patient [68,69]. Dennis et al. used a combination of global gene expression, synovial histology, and cellular analyses to identify pre-treatment serum biomarkers which predict response to anti-TNF therapy [42]. Based on synovial gene expression, they describe the four synovial pathotypes: lymphoid, myeloid, low inflammatory, and fibroid. They observed differences in the number of B cells in each phenotype with high numbers in the lymphoid and myeloid groups and an absence of B cells in the low inflammatory and fibroid groups. Two serum biomarkers were identified which reflected two different synovial phenotypes: sICAM1 titers reflected the myeloid phenotype, and the chemokine CXCL13 reflected the lymphoid phenotype. Finally, the group found that sICAM1^high^CXCL13^low^, i.e., the myeloid pathotype, had higher responses to anti-TNF inhibition with adalimumab, whereas sICAM1^low^CXCL13^high^ lymphoid pathotype patients had the largest response rates to tocilizumab.

The COMBINE study used multi-omics data (DNA, RNA, proteomics, flow cytometry) to find predictors of anti-TNF response, and found that a combination could explain 51% of the variation in DAS28 with an AUC of 0.815 [70], although such approaches are not currently feasible on a large scale. The findings require independent validation as the model is likely to be over-fitted, having been derived from a single dataset.

As synovial biopsy becomes more readily performed and accepted by patients, transcriptomics of specific cell types is being performed to build up a transcriptional profile of synovial macrophages [71]. It is likely that such cutting-edge technologies will be integrated with clinical and immune biomarkers to further develop prediction models, and this is the subject of several on-going studies including the MATURA program in the UK [72] and the Accelerated Medicines Partnership RA/SLE Network in the US [73].

## 4. Conclusions

New treatments for rheumatoid arthritis and other autoimmune diseases are now targeted towards specific components of the immune system. Whereas, monoclonal antibodies are highly specific for their immune targets, SMIs such as JAK-STAT inhibitors target intracellular signaling downstream of cytokine receptors [74]. Immune cells have a central role in autoimmune pathology, and given that they are also targets of these treatments, investigations of immunophenotyping and treatment response are all the more important. The exploration of serum proteins, for example, cytokines, has not so far been fruitful in revealing treatment response biomarkers [75]. Other disciplines have utilised functional immunology to identify diagnostic or theragnostic biomarkers. For example, the IFNγ-release assay (IGRA) uses tuberculosis antigens to stimulate CD4^+^ T cells and measures the level of IFNγ release to determine previous tuberculosis exposure [76].

Currently, immunophenotyping is not yet useful in clinical practice to predict response to therapy in any discipline, although intense research is ongoing in the fields of oncology [77], hematology [78], rheumatology [79], and transplant medicine [80]. Developing reproducible methods of immune cell phenotyping is therefore crucial to both understanding mechanisms of disease and to discovering cellular immune biomarkers to predict treatment response. We feel that the future of treatment prediction lies in the inclusion of a number of cellular and serum biomarkers, which may or may not include functional immunological assays, combined with recognised clinical predictors of response, into a multivariate prediction model for each immune-mediated disease. High-throughput methods of measuring each of these variables will only make such models more accurate in their prediction of response.

## Figures and Tables

**Figure 1 jpm-09-00046-f001:**
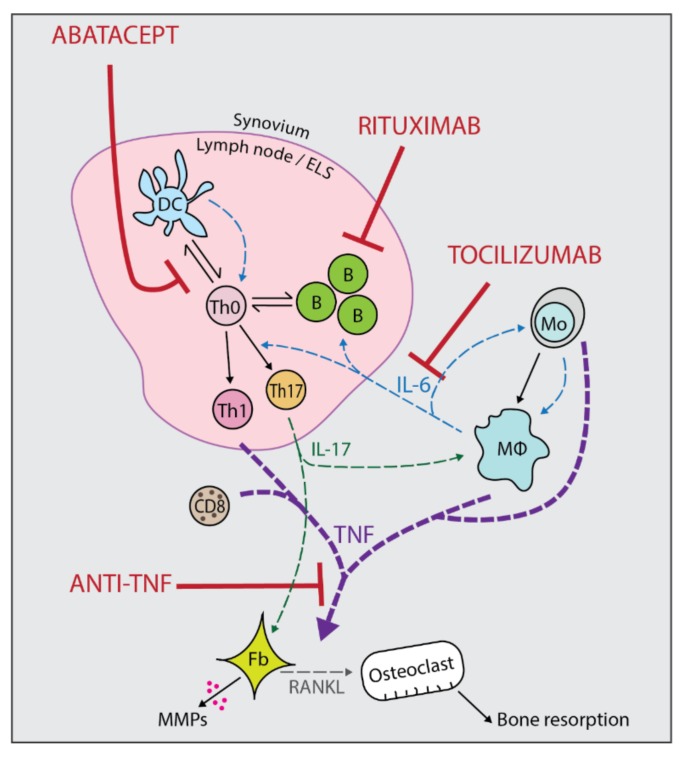
Sites of action of the main biologic drugs used in rheumatoid arthritis (RA). The cartoon represents the synovial membrane compartment with an associated ectopic lymphoid structure. Cytokine pathways are shown by dashed arrow lines. Abatacept works by blocking co-stimulation between DCs and T cells. Rituximab depletes B cells which may present antigen to T cells and produce autoantibodies. Tocilizumab blocks the actions of IL-6 which has many pro-inflammatory effects within the synovium. Anti-TNF blocks the actions of TNF which is able to activate fibroblasts and osteoclasts resulting in joint damage. B: B cell, CD8: Cytotoxic T cell, DC: Dendritic cell, ELS: Ectopic lymphoid structure, Fb: Fibroblast, MΦ: Macrophage, Mo: Monocyte, T: T cell, Th0: Naïve T helper cell, Th1: T helper 1 cell, Th17: T helper 17 cell.

**Table 1 jpm-09-00046-t001:** Summary of the main studies investigating immune biomarkers as predictors of treatment response.

Biomarker	Drug	Ethnicity	Concurrent DMARDs	Response Criteria	Predictor of Response	*N* Cases	Reference
Seropositive status	Infliximab	Caucasian	100%	DAS28 ≥ 1.2	Low ACPA titre predictive of response (PPV 0.95)	30	[23]
		Caucasian	73%	EULAR	Low RF/ACPA titre	1195	[24]
	Infliximab	Asian	100%	ΔCRP	Low RF titre	62	[25]
		Caucasian	100%	EULAR	ACPA not associated with response	42	[26]
		Caucasian	100%	DAS28	ACPA not associated with response	31	[27]
		Mixed	n/a	DAS28 ACR20 EULAR	Meta-analysis found no association between seropositive status and anti-TNF response	5561	[28]
	Adalimumab	Caucasian	100%	EULAR DAS28 ACR20	ACPA^+^	245	[29]
	Infliximab	Asian	100%	DAS28	High RF/ACPA titre	307	[30]
		Caucasian	n/a	CDAI	No association	1715	[31]
	Abatacept	Caucasian	64.8%	EULAR	ACPA^+^ (OR 1.9; 1.2–2.9)	558	[32]
		Caucasian	100%	EULAR DAS28 ACR20	High ACPA titre	252	[29]
		Caucasian	n/a		Higher continuation of abatacept in seropositive cohorts	1357	[33]
		Caucasian	n/a	CDAI	ACPA^+^	566	[31]
		Caucasian	75%	Retention rate	Double RF^+^/ACPA^+^	2350	[34]
	Rituximab	Mixed	n/a	ACR20 EULAR	Meta-analysis found RF^+^ associated with treatment response	2103	[35]
		Caucasian	n/a	DAS28	Meta-analysis showing seropositive patients respond better to rituximab than seronegative patients	2177	[36]
		Caucasian	74.6%	EULAR DAS28	High ACPA titre	114	[37]
	Tocilizumab	Mixed	n/a	ACR20 EULAR	Meta-analysis found RF^+^ associated with treatment response		[35]
IL-6	Etanercept	Asian	n/a	n/a	Increased IL-6 (with low survivin) associated with response (OR 19.7, CI 4.1–94.8)	73	[38]
	Tocilizumab	Caucasian	48.6%	EULAR	Increased IL-6 (with low IL-6R) associated with response	63	[39]
IL-33	Rituximab	Caucasian	100%	EULAR	High IL-33 (and ACPA^+^) associated with response (OR 29.61, CI 1.3–674.8)	74	[40]
CXCL13	Anti-TNFs	Caucasian	100%	EULAR	High CXCL13 (and high CXCL10) associated with response (AUC 0.83)	29	[41]
	Tocilizumab	Caucasian	0%	ACR	High CXCL13 (with low sICAM1) (AUC 0.65)	198	[42]
CCL19	Rituximab	Caucasian	100%	EULAR	High CCL19 associated with response (OR 1.43, CI 1.08–1.90)	208	[43]
B cells	Anti-TNFs	Caucasian	69%	EULAR	High CD27^+^ B cells associated with response (RR 4.9, CI 1.3–18.6)	21	[44]
	Abatacept	Caucasian	51.2%	EULAR	High CD27^+^ and/or CD38^+^ B cells associated with response	43	[45]
	Rituximab	Caucasian	100%	EULAR	High CD27^−^ B cells are associated with response	154	[46]
CD8^+^ T cells	Etanercept	Caucasian	n/a	EULAR	High apoptotic epitope-specific CD8^+^ T cells associated with response (AUC 0.82)	16	[47]
	Abatacept	Caucasian	n/a	DAS28	Low CD28^−^ CD8^+^ T cells is associated with response	32	[48]
NK cells	Tocilizumab	Caucasian	60%	DAS28	Low CD56^bright^CD16^−^ NK cells associated with response	20	[49]
Type I interferon signature	Anti-TNF	Hispanic	71–100%	EULAR	High type I IFN activity associated with response (OR 1.36, CI 1.05–3.29)	35	[50]
	Rituximab	Caucasian	55%	EULAR	High type I IFN signature negatively associated with response	20	[51]
		Caucasian	77%	DAS28	High type I IFN signature negatively associated with response (AUC 0.87)	26	[52]

Table 1 outlines the main studies investigating immune biomarkers predicting treatment response in RA. Immune signatures are listed alongside the biologic drug studied, the ethnicity of the patient group, the percentage of the cohort taking concurrent disease-modifying anti-rheumatic drugs (DMARDs), the outcome measure used, the main findings, and the sample size. ACPA: Anti-citrullinated peptide antibody. ACR: American College of Rheumatology [53]. AUC: Area under the curve. CDAI: Clinical disease activity index. CI: Confidence interval. DAS28: Disease activity score in 28 joints [54]. EULAR: European League Against Rheumatism [55]. OR: Odds ratio. PPV: Positive predictive value. RF: Rheumatoid factor.

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
