# Peer review of "Using the Immunophenotype to Predict Response to Biologic Drugs in Rheumatoid Arthritis"

_jpm, 2019, doi:10.3390/jpm9040046_

Round 1

Reviewer 1 Report

The present paper is an easily readable revision of current evidence about laboratory predictors of response to biological therapies in rheumatoid arthritis. The table summarizes all the studies now available on the topic.

In the Introduction, the Authors mention the cost issue; it could be useful to cite also the introduction of biosimilar drugs as a possible response to this issue.

In the paragraph about Rituximab, the Authors cite hypogammaglobulinemia as the primary security issue, however, must be taken into account also the risks of neutropenia and the possible development of fatal brain infection in the patients without any evidence of humoral or cellular secondary immunodeficiency.

This review is not intended only to rheumatologists so the Authors could insert some more explanations about the response criteria cited in the table; alternatively, they could add a reference explaining these criteria.

The meaning of ACPA and RF is clarified only in the main text; however these acronyms are cited in the table that is presented before in the paper.

The review is written in a correct English however corrections are needed, especially regarding the use of commas (e.g., line 73: ... antigen to T cells, and produce...; line 89: ... causing systemic inflammation, including fever, high CRP, and fatigue.)

Author Response

The present paper is an easily readable revision of current evidence about laboratory predictors of response to biological therapies in rheumatoid arthritis. The table summarizes all the studies now available on the topic.

Many thanks for your review. I will supply replies below each of your points in bold.

In the Introduction, the Authors mention the cost issue; it could be useful to cite also the introduction of biosimilar drugs as a possible response to this issue.

I have amended the following sentence: "and biosimilars which are the biologic equivalent of generic medicines are continually driving down the cost of biologic drugs." 

In the paragraph about Rituximab, the Authors cite hypogammaglobulinemia as the primary security issue, however, must be taken into account also the risks of neutropenia and the possible development of fatal brain infection in the patients without any evidence of humoral or cellular secondary immunodeficiency.

Thank you. I have amended the sentence: "The main safety considerations are the development of hypogammaglobulinaemia and neutropaenia which predispose to serious infections, and the reactivation of a number of fatal brain infections such as JC virus."

This review is not intended only to rheumatologists so the Authors could insert some more explanations about the response criteria cited in the table; alternatively, they could add a reference explaining these criteria.

Thanks for pointing this out. I have expanded on these response criteria more fully in the table legend and included the relevant references.

The meaning of ACPA and RF is clarified only in the main text; however these acronyms are cited in the table that is presented before in the paper.

Thank you. I have amended this in the table legend.

The review is written in a correct English however corrections are needed, especially regarding the use of commas (e.g., line 73: ... antigen to T cells, and produce...; line 89: ... causing systemic inflammation, including fever, high CRP, and fatigue.)

Thank you. Although the use of commas is subjective and may or may not improve the reading style without changing the meaning of a sentence, I have reviewed all the commas in this manuscript and attempted to improve the reading style accordingly.

Reviewer 2 Report

Mulhearn et al. provide a well-written narrative on the current therapies in RA, focusing on predictors of response to biologics. The paper reads well and puts forward ideas that may be of interest to other rheumatologists.

I would advise revising the conclusions, they should be a short summary of the main messages from your paper, without drawing back to several  references. The latter can be included in a prior fragment, which could describe future prospects and the authors expert opinion on the present research. On a side note, the concept of prediction models, rather than a singular marker, seems to be a promising view from the future, particularly considering the wide heterogeneity of RA.

58 - As JAK inhibitors do not belong to biologics, but rather tsDMARDs, I would omit the statement "not strictly biologic".

Regarding section 3.1 - It would be interesting to comment on the role of very early RA (VERA) with regard to predicting response.

Table 1, reg. ref 22 - Did this study investigate multiple TNFs or just IFX?

Author Response

Mulhearn et al. provide a well-written narrative on the current therapies in RA, focusing on predictors of response to biologics. The paper reads well and puts forward ideas that may be of interest to other rheumatologists.

Thank you for your review. I reply reply to each point below in bold.

I would advise revising the conclusions, they should be a short summary of the main messages from your paper, without drawing back to several references. The latter can be included in a prior fragment, which could describe future prospects and the authors expert opinion on the present research. On a side note, the concept of prediction models, rather than a singular marker, seems to be a promising view from the future, particularly considering the wide heterogeneity of RA.

Thank you. We have attempted to break the conclusion down according to your suggestions and include the importance of development of a prediction model as you have pointed out.

58 - As JAK inhibitors do not belong to biologics, but rather tsDMARDs, I would omit the statement "not strictly biologic".

Thank you. I will clarify this in the text.

Regarding section 3.1 - It would be interesting to comment on the role of very early RA (VERA) with regard to predicting response.

I agree that very early rheumatoid arthritis would be interesting to comment on and I have included the following sentence and reference in the introduction (1.): "It may even be possible to shut down the disease process entirely in very early rheumatoid arthritis (VERA) which is defined as less than 3 months of disease activity [5].

Table 1, reg. ref 22 - Did this study investigate multiple TNFs or just IFX?

Thank you. This appears ambiguous in the table and I have corrected this now in the table as IFX only.